# Different Dose of Sucrose Consumption Divergently Influences Gut Microbiota and PPAR-γ/MAPK/NF-κB Pathway in DSS-Induced Colitis Mice

**DOI:** 10.3390/nu14132765

**Published:** 2022-07-04

**Authors:** Xuejiao Zhang, Bowei Zhang, Bo Peng, Jin Wang, Yaozhong Hu, Ruican Wang, Shuo Wang

**Affiliations:** 1Tianjin Key Laboratory of Food Science and Health, School of Medicine, Nankai University, Tianjin 300071, China; xuejiaozhang@mail.nankai.edu.cn (X.Z.); bwzhang@nankai.edu.cn (B.Z.); iq1226jsnpb@hotmail.com (B.P.); wangjin@nankai.edu.cn (J.W.); yzhu@nankai.edu.cn (Y.H.); rcwang@nankai.edu.cn (R.W.); 2College of Food Science and Technology, Hebei Agricultural University, Baoding 071000, China; 3Institute of Public Health, Nankai University, Tianjin 300071, China

**Keywords:** sucrose, gut microbiota, SCFAs, PPAR-γ, MAPK/NF-κB

## Abstract

Sugar reduction and sugar control are advocated and gaining popularity around the world. Sucrose, as the widely consumed ingredient in our daily diet, has been reported a relation to gastrointestinal diseases. However, the role of sucrose in inflammatory bowel disease remains controversial. Hence, our study aimed to elucidate the potential role of three doses of sucrose on DSS-induced colitis in C57BL/6 mice and the underlying mechanisms. The results showed that low-dose sucrose intervention alleviated colitis in mice, reducing the expression of inflammatory cytokines and repairing mucosal damages. In contrast, high-dose sucrose intervention exacerbated colitis. Furthermore, three doses of sucrose administration markedly altered gut microbiota composition. Notably, the low-dose sucrose restored microbial dysfunction and enhanced the production of short chain fatty acids (SCFAs). Specifically, the abundance of SCFAs-producing bacteria *Faecalibaculum*, *Bacteroides,* and *Romboutsia* were increased significantly in the LOW group. Consistently, PPAR-γ, activated by SCFAs, was elevated in the LOW group, thereby inhibiting the MAPK/NF-κB pathway. Together, our study demonstrates the differential effects of sucrose on colitis at different doses, providing a scientific basis for measuring and modifying the safe intake level of sugar and providing favorable evidence for implementing sugar reduction policies.

## 1. Introduction

Sucrose is a nutritional sweetener with wide application and high accessibility, and the global average consumption level continues to maintain a high level. Given the rising global burden of high sugar-related metabolic diseases, sugar has come into spotlight as a new critical public health problem [1,2]. Moreover, there was a growing interest in a policy intervention to reduce the consumption of sugar-sweetened beverages at a population level [3]. Previous studies have suggested that the health harms were closely proportional to the dose of sugar [4,5]. However, the association between sucrose intake and intestinal disease remains controversial [6]. Despite the fact that several clinical cohort studies have observed increased sugar consumption in Crohn’s disease (CD) patients [7,8], there are null associations between added sugars and inflammatory bowel disease (IBD) [9,10]. In some interventional studies, excessive sucrose intake promoted the progression of intestinal disease in mice [11,12,13]. However, clinical evidence has shown that adherence to a low-sucrose diet can reduce gastrointestinal symptoms in patients with irritable bowel [14]. It is thus significant to clarify the relationship between sucrose intake and body health for the rational formulation of sugar reduction policies.

Inflammatory bowel disease (IBD), mainly composed of Crohn’s disease (CD) and ulcerative colitis (UC), is an intestinal disease with a global epidemic [15]. The site of IBD often involves the colon and ileum, so patients generally suffer from diarrhea, abdominal pain, and bloody stool due to the recurrence of the disease. It is speculated that the incidence of IBD is possibly related to diet and intestinal homeostasis based on the evidence of epidemiological investigation and original research, although current knowledge of etiology is still controversial [16,17,18]. This speculation highlights the necessity of a dietary guidance provided to doctors and patients regarding safety, harmfulness, and recommended dosage of food.

Association between gut microbiota and the establishment of IBD has been identified over the past years [19,20,21]. Many studies have shown that IBD can be prevented/attenuated under germ-free condition, indicating that microorganisms play crucial roles in the development of IBD [22,23]. Alterations in the composition of gut microbiome have been reported in previous IBD studies. The interaction between intestinal microbiota and IBD is by the alterations in the microbiota composition and metabolites profiles [24]. Studies have demonstrated that bile acid derivatives, short chain fatty acids (SCFAs), and tryptophan metabolites have become the focus of research due to the close relationship with IBD [25,26,27,28]. Dietary nutrients can profoundly change the composition of microbiota composition and metabolites profiles [29,30]. However, there is limited understanding of how sucrose intake affects microbial homeostasis.

Peroxisome proliferator-activated receptor-γ (PPAR-γ) expression was reported to decrease in UC patients during disease activity [31]. Evidence suggests that PPAR-γ is a critical regulator during intestinal inflammation, interfering with transcription factors involved in the inflammatory response. PPAR-γ decreased expression may promote the production of inflammatory mediators, resulting in the activation of nuclear factor kappa B (NF-κB) and mitogen-activated protein kinases (MAPK) pathways [32,33]. In addition, another key role of PPAR-γ in regulating intestinal inflammation has been proposed. PPAR-γ signaling activated by microbial metabolites butyrate is a homeostatic pathway by promoting the energy metabolism of colonic epithelium towards β-oxidation to maintain intestinal hypoxia and prevent the expansion of pathogenic bacteria [34].

In this study, influences of three doses of sucrose on weight, diarrhea, colon histomorphology, inflammatory cytokines, gut microbial composition, and SCFAs production changes in DSS-induced colitis mice were evaluated systematically. We demonstrated that different doses of sucrose intake altered the disease severity and microbial homeostasis differently. Of note, low dose sucrose intake relieved the colonic inflammation and amended the gut dysbiosis. Meanwhile, excess sucrose consumption disturbed microbial homeostasis and exacerbated the inflammation. Mechanistically, the enhancement of PPAR-γ related signaling pathways activated by SCFAs was involved in the remission of colitis. Low dose sucrose intake exhibited a preventive effect on colitis, further suggesting the importance of sugar control. We believe that the results of this study provide a new perspective on the safe amount of sugar.

## 2. Materials and Methods

### 2.1. Animals and Experimental Design

All animals’ experiments were conducted following the protocols approved by the Nankai animal resources center (SYXK-2019-0001). All animals were kept in specific pathogen-free conditions (12 h light and dark cycle, 22 ± 2 °C, humidity 50 ± 10%). All animals had free access to a control chow and water.

The C57BL/6 mice (6 weeks) were randomly divided after 7 days of acclimation. Each of the 5 groups was formed composing of 10 mice: Control group (CON group), Dextran sodium sulfate modeling group (DSS group), DSS + 7.5 mg/mL Sucrose group (LOW group), DSS + 15 mg/mL Sucrose group (MID group), and DSS + 30 mg/mL Sucrose group (HI group). In order to correspond to the free sugar intake of 25 g recommended by the World Health Organization, the amount of sucrose used in the MID group was obtained according to the following calculation:$$\frac{25\;g\;day^{-1}60\;kg^{-1}\times \mathrm{average}\;\mathrm{mouse}\;\mathrm{weight}\;0.02\;kg\times \mathrm{Mehh}\;\mathrm{Rubner}\;\mathrm{coefficient}\;9}{\mathrm{average}\;\mathrm{daily}\;\mathrm{water}\;\mathrm{intake}\;5mL}\;=15\;mg\;mL^{-1}day^{-1}\;MID\;group$$

Sucrose (# S112226, Shanghai Aladdin Biochemical Technology Co., Ltd., Shanghai, China) was dissolved in water according to the specified dose and given to mice for 21 days. The colitis model was obtained by 2.5% (*w/v*) dextran sulfate sodium (DSS, MW: 36, 000-50, 000 Da, MP Biomedicals, Solon, OH, USA) dissolving in drinking water for 7 days. The drinking water was changed every day. The disease activity index (DAI) of every experimental animal was monitored simultaneously (Appendix A). Then, the result of DAI score was gained as the average of the three parameters according to the rules previously described [35]. The mice were sacrificed on day 28, and the tissues were collected for analysis.

### 2.2. Histological Analysis

The distal colonic section was obtained 1–2 cm away from the anus. The tissues were soaked in 10% formalin overnight for denaturation and fixation. Then, 5 µm colon sections were embedded in paraffin for hematoxylin and eosin (H&E) staining.

### 2.3. Quantitative Reverse Transcription PCR (RT-qPCR)

An amount of 1 mL Trizol (#15596018, Invitrogen, Waltham, MA, USA) was applied to extract all RNA from 30–50 mg colon tissues. RNA quantification was measured by NanoPhotometer^@^ N50 (Implen Gmbh Co., Ltd., Munich, Germany). Then, the RevertAid First Strand cDNA Synthesis Kit (#k1622, Thermo Scientific™, Waltham, MA, USA) was used for reverse transcription of 1 μg RNA. qPCR was performed by utilizing the SYBR green master mix (#M3003, Vazyme, Nanjing, China) following the manufacturer’s protocol. The target gene’s relative expression level was quantified by the 2^−ΔΔCt^ method and standardized to the expression of the housekeeping gene β-actin. The primers were listed in Appendix A.

### 2.4. Enzyme-Linked Immunosorbent Assay (ELISA)

Cytokine levels of IL-1β, TNF-α, and IL-6 in serum were determined by the ELISA kits (#CK-E20533M, #CK-E20220M, #CK-E20012M, Biocalvin, Suzhou, China) in accordance with the manufacturer’s protocol. 

### 2.5. Immunofluorescence

An amount of 4% paraformaldehyde solution was used for colonic tissue fixation. Then, the prepared colon tissue was used for immunofluorescence staining of PPAR-γ. The colon sections were incubated with PPAR-γ primary antibody (#C26H12, 1:100, Cell Signaling Technology, Danvers, MA, USA) at 4 °C overnight after blocked with 5% BSA at room temperature for 30 min. Subsequently, the slices were incubated with the goat anti-mouse secondary antibody (#ZF-0313, 1:100, ZS-Bio, Evanston, IL, USA) for 30 min at 37 °C. Anti-fluorescence attenuation sealants containing DPAI (#S2110, Solarbio, Beijing, China) were used for sealing and nuclear staining.

### 2.6. Western Blotting

RIPA lysis buffer (#P0013B, Beyotime, Haimen, China) with Halt™ Protease Inhibitor Cocktail (100×) (#78430, Thermo Scientific™) was applied to the acquisition of total proteins in colon tissue. The concentration of the protein was obtained though Detergent Compatible Bradford Protein Assay Kit (#P0006C, Beyotime, Shanghai, China) according to the manual. Then, 30 μg protein and the 5× loading buffer were mixed and boiled at 95 °C for 10 min. Then, 40 μg of protein were separated by sodium dodecyl sulfate-polyacrylamide gel electrophoresis gel (SDS-PAGE). The results were then blotted onto the nitrocellulose filter membrane (NC) by wet-transferred process. After being sealed with 5% skimmed milk powder at room temperature for 1 h, the membrane was incubated with primary antibody overnight at 4 °C. The antibodies involved were as follows: ERK1/2 (42/44 kDa, 1:2000, #R22685, Zen Bioscience, Durham, NC, USA), p-ERK1/ERK2 (41/43 kDa, 1:1000, #301245, Zen Bioscience), JNK1/2/3 (48/54 kDa, 1:1000, #R24780, Zen Bioscience), p-JNK1/2/3 (46/54 kDa, 1:1000, #381100, Zen Bioscience), p38 (41 kDa, 1:1000, #R25239, Zen Bioscience), p-p38 (43 kDa, 1:500, #310091, Zen Bioscience), iκb (36–39 kDa, 1:1000, #383322, Zen Bioscience), p-iκb (34 kDa, 1:1000, #340776, Zen Bioscience), p65 (65 kDa, 1:1000, #8242, Cell Signaling Technology, Danvers, MA, USA), and p-p65 (65 kDa, 1:1000, #3033, Cell Signaling Technology). After washing, the membranes were incubated with a suitable secondary antibody. The internal reference protein β-Actin was used to homogenize results. The quantification of protein was displayed by the Pierce™ ECL Western Blotting Substrate (#32106, Thermo Fisher, Waltham, MA, USA) following the manufacturer’s instructions. The Bio-Rad ChemiDoc^TM^ MP Imaging System was used for image acquisition. The intensity measurements of the bands were performed using ImageJ.

### 2.7. Short Chain Fatty Acids Determination

Before testing, feces from mice were collected and stored at −80 °C refrigerators. Feces were treated with sulfuric acid and ether after crushing and homogenization. The ether phase was obtained after centrifugation and passing through the organic phase filter membrane. Then, the content of SCFAs were obtained by gas chromatography (Agilent 7890A, DB-FFAP capillary column, Agilent Technologies, Santa Clara, CA, USA)

### 2.8. 16 s rRNA Gene Sequencing and Analysis

The total DNA in the colonic contents was extracted and quantified for PCR amplification. The V3-V4 region of 16S rRNA gene was amplified by universal primers 338F(5′-ACTCCTACGGGAGGCAGCA-3′) and 806R(5′-GGACTACHVGGGTWTCTAAT-3′). Sequencing analysis was performed using Illumina Hiseq 2500 system (PE250). Beijing Biomarker Technologies Co., Ltd.( Beijing, China) provided technical support. 

### 2.9. Statistical Analysis

All statistical analyses were performed and visualized by GraphPad Prism 8.0. Results were presented as the mean ± standard deviations of the mean (SEM). Significances of different groups were performed by one-way ANOVA or two-way ANOVA followed by Tukey’s post hoc test. P values less than 0.05 (*p* < 0.05) were indicated to be significant in all analyses. 

## 3. Results

### 3.1. Effects of Different Dosage Sucrose on Symptoms of Colitis in Mice

The pharmacodynamics of the three different dosages of sucrose were explored on DSS-induced colitis mice following the workflow illustrated in Figure 1A. It could be concluded from Figure 1B that there was no significant difference in oral glucose tolerance among the groups after 3 weeks intervention of different doses of sucrose. Strikingly, compared to the MID group, the LOW group exhibited a better effect in protecting the colitis, while the HI group exhibited an aggravating symptom of colitis after 7 days of DSS treatment. As shown in Figure 1C, the LOW group slowed down the weight loss of mice, the MID group showed a similar result to the DSS group, while the HI group lost more bodyweight than the DSS group. Meanwhile, similar results were also found in the DAI score and colon length. The DAI score was remarkably increased upon the intake of sucrose in a dose-dependent manner (Figure 1D). The shortening of the colon was alleviated in the LOW group and aggravated in the HI group compared with the DSS group (Figure 1E–F). Based on the observations, we can declare that low-dose sucrose intervention delayed the progression of colitis in mice.

### 3.2. Effects of Different Dosage Sucrose on Colon Inflammation

To further understand the effects of sucrose in colitis, the gene expression levels of inflammatory cytokines in the colon were measured. There was a twofold decrease in the expression of IL-6 and TNF-α, and a 1.5-fold decline of IL-1β in the LOW group compared to the DSS group. Similar levels of inflammatory cytokines expression were observed in the MID group. On the contrary, the expression of IL-6 and TNF-α increased 3 times and 2.5 times in the HI group, respectively (Figure 2A–C). In order to further confirm the change of sucrose on inflammation, the protein levels of inflammatory cytokines in serum were determined by ELISA (Figure 2D–F). In particular, the LOW group significantly reduced the production of inflammatory factors compared to the DSS group. Not surprisingly, the result of the MID group was more similar to that of the LOW group. The HI group aggravated the inflammation compared to the colitis model group. The above results suggested that low-dose sucrose intake might have a preventive effect on the inflammation of colitis.

### 3.3. Effects of Different Dosage Sucrose on Mucosal Barrier

In order to further evaluate the pathological injury of the colon, HE staining was performed. The result revealed that the LOW group alleviated the histological damage caused by DSS, which was characterized by the thickening of the intestinal mucosa, epithelium, and mucosal muscle layer, the improvement of recess and gland loss, and the reduction of inflammatory cell infiltration (Figure 2G). In addition, an opposite result, the aggravation of mucosal injury, was found in the HI group. Consistently, the gene expression levels of tight junction protein ZO-1 and Occludin in colon tissue were significantly higher in the LOW group than the HI group (Figure 2H,J). However, there was no significant change in the expression of Claudin (Figure 2I). Together, these observations indicated that low-dose sucrose prevented the progression of inflammation and improved mucosal damage. 

### 3.4. Effects of Different Dosage Sucrose on the Gut Microbiota Composition

To further observe the role of different doses of sucrose interventions, the composition of gut microbiota from colonic contents was analyzed. The Chao and Simpson indices were detected to represent the α-diversity for each group (Figure 3A). Compared with the colitis model group, different doses of sucrose intervention reduced the Chao index representing the species’ riches. In addition, the lowest Chao index was observed in the HI group. The uniformity of microorganisms characterized by the Simpson index significantly decreased in the LOW group compared to that in other groups. 

The aforementioned results revealed that low-dose sucrose enhanced the diversity of gut microbiota, while high-dose sucrose was the opposite. The β-diversity of the gut microbiome was demonstrated through principal coordinates analysis (PCoA) between OTUs, and analysis of similarities (ANOSIM) based on the Binary_jaccard distance (Figure 3B). The results suggested that different doses of sucrose changed the composition of microorganisms after 3 weeks of intervention (ANOSIM: R = 0.606, *p* = 0.001). The phylum and genus levels of microbial composition were defined (Figure 3C–E). The abundance of *Acidobacteria*, *Bacteroidetes*, and *Actinobacteria* increased in the LOW group compared to the DSS group. Yet, the abundance of *Firmicutes* and *Verrucomicrobia* decreased. The composition of the MID group was similar to the HI group. The abundance of *Firmicutes*, *Bacteroidetes*, *Cyanobacteria*, and *Patescibacteria* was higher than the DSS group. At the genus level, the heat map of species with the abundance top 15 reconfirmed that different doses of sucrose changed the composition of microorganisms. The abundance of *Bacteroides* and *Faecalibaculum* was significantly increased in the LOW group compared with the DSS group. Interestingly, significant suppression of *Lachnospiraceae* and *Muribaculaceae* was found in the LOW group compared to the DSS group. Together, low dosage sucrose administration did cause an alteration in the gut microbiota composition.

### 3.5. Effects of Different Dosage Sucrose on the Microbial Metabolism Pathways

The functional gene difference of microbial communities among different groups was obtained under the KEGG metabolic pathway (level 3) by PICRUSt analysis (Figure 4A). There were five different metabolic pathways identified between the LOW group and the DSS group. Moreover, the benzoate degradation pathway was upregulated in the HI group compared to the DSS group. It was worth noting that low-dose sucrose intervention upregulated the metabolic pathway and starch and sucrose metabolism pathway, which were closely related to the production of SCFAs. The distinct difference between low-dose and high-dose sucrose intervention on colitis attracted great attention. Through the results, it can be found that the LOW group was more inclined to the sucrose metabolism, glycolysis/gluconeogenesis pathways. At the same time, there was an enrichment of amino sugar and nucleotide sugar metabolism pathways. In contrast, high-dose sucrose intervention upregulated the pathways such as biosynthesis of secondary metabolites, aminoacyl-tRNA biosynthesis, two-component system, and biosynthesis of amino acids. The opposite effect of low-dose and high-dose sucrose intake on colitis may be caused by other interference with the metabolic homeostasis of the microbiome. Low-dose sucrose intervention reversed the dysfunction of the microbial metabolic pathway and improved the functional potentials in environmental sensing, which led to the maintenance of colonic homeostasis.

### 3.6. Effects of Different Dosage Sucrose on the SCFAs Production

Representative species of each group were identified by line discriminant analysis effect size (LEfSe) at the taxonomic level from phylum to genus. As shown in Figure 4B, *Lachnospiraceae*, *Akkermansia,* and *Lactobacillus* were dominant in the colitis model group. The abundance of *Lachnospiraceae_NK4A136_group*, *Ruminococcaceae,* and *Rhodospirillales* was greater in the HI group than in the DSS group. Notably, the abundance of *Erysipelotrichaceae*, *Bifidobacterium*, *Faecalibaculum*, *Bacteroides*, and *Romboutsia* were increased significantly in the LOW group compared to the DSS group. Based on previous studies, these bacteria were involved in the degradation of carbohydrates and the production of SCFAs [36,37]. Hence, the contents of five SCFAs in colonic contents were determined using gas chromatography (Figure 5A–F). As expected, the levels of acetic acid, propionic acid, butyric acid, isovaleric acid, valeric acid, and total SCFAs markedly increased in the LOW group compared to the DSS group. Interestingly, the results of the MID group and the HI group were similar, consistent with the similar composition of gut microbiome.

### 3.7. Effects of Different Dosage Sucrose on PPAR-γ and MAPK/NF-κB Pathway

It was well known that SCFAs, as the endogenous ligands for PPAR-γ, could bind and active PPAR-γ. Previous studies have shown that depletion of butyrate-producing microbes reduced epithelial signaling through the intracellular butyrate sensor PPAR-γ [38]. To further explore the mechanism of low-dose sucrose in mitigating colitis, the expression of PPAR-γ in the colon was detected at the gene level by qPCR and protein level by immunofluorescence (Figure 6A,B). The results showed that the expression of PPAR-γ was negatively correlated with sucrose dose. The expression level of PPAR-γ in the LOW group increased significantly compared to the DSS group. MAPK and NF-κB pathways have been reported to be activated and hyperphosphorylated in IBD patients [39]. At the same time, a close cascade existed between PPAR-γ and the MAPK/NF-κB pathway [40]. Consequently, we speculated that the remission of the LOW group might be mediated through the MAPK/NF-κB pathway. Therefore, the MAPK/NF-κB signaling pathway was determined by Western blot, containing analysis of total ERK, JNK, and p38 protein expression in the MAPK pathway, total iκb and p65 protein expression in the NF-κB pathway, as well as their phosphorylation levels (Figure 6C,D). As predicted, the MAPK/NF-κB pathway was strongly inhibited in the LOW group with high-level expression of PPAR-γ. In addition, protein phosphorylation levels were higher in the HI group than in the DSS group. Comparable results were observed in the MID group and the DSS group. To further observe the critical role of the gut microbiome, Spearman correlation analysis at the genus level was performed (Figure 7). The result revealed that the SCFAs levels were correlated with the abundance of *Faecalibaculum*, *Romboutsia*, *Bacteroides*, and *Turicibacter*. Crucially, *Faecalibaculum* was most strongly positively correlated with the PPAR-γ and SCFAs levels but negatively correlated with the IL-6, IL-1β, and TNF-α levels in the colon. Moreover, the IL-6 and IL-1β levels showed that significantly negative correlations with *Bacteroides* and *Romboutsia*. *Lachnospiraceae_NK4A136* and *Ruminococcaceae*, the specific biomarkers in the HI group, were negatively correlated with PPAR-γ and SCFAs levels, especially butyric acid levels. Together, the aforementioned results confirmed our hypothesis that low-dose sucrose intervention alleviated DSS-induced colitis in mice by restoring microbial dysfunction, promoting the production of SCFAs, and activating the PPAR-γ/MAPK/NF-κB signaling pathway.

## 4. Discussion

Over the past decade, several epidemiological studies have observed the close association between diet and IBD [41,42]. The high-sucrose diet has been linked to triggering IBD [43]. A previous study has found that diet rich in sucrose promoted pro-inflammatory response via gut microbiota in mice [13]. Collectively, sucrose has become synonymous with harmful nutrients. However, conflicting results have also confirmed that the occurrence of IBD is not associated with sucrose intake [44]. Therefore, as an essential component of the daily diet, the safe intake of sucrose has become an urgent problem to be solved. In our study, three doses of sucrose (according to the recommended intake of 25 g proposed by the WHO) were applied to animals modeling colitis for 3 weeks to explore the effects of sucrose intervention on colitis. Our results demonstrated that excessive sucrose intake exacerbates colitis, and that reducing sucrose intake reduces its harm and has a health effect that improves inflammation. Moreover, low-dose sucrose prevented gut microbial disorder, promoted the production of SCFAs, and activated the PPAR-γ/MAPK/NF-κB signaling pathway. Thus, our results suggest that although excessive sucrose can lead to dysfunction, low-dose sucrose is feasible for maintaining healthy homeostasis, at least for colitis in mice.

DSS-induced colitis modeled successfully was usually characterized by a series of symptoms of colitis including weight loss, bloody stool, and colon shortening. Simultaneously, the expression of inflammatory cytokines in the colon rises sharply, accompanied by the injury of colonic mucosa. Therefore, these can be targeted for the remission of colitis [45]. In this study, weight loss, DAI score rising, and colon shortening were delayed under a low-dose sucrose consumption. Compared to the DSS group, the expressions of inflammatory cytokines (IL-6, IL-1β, and TNF-α) were suppressed in the LOW group and increased in the HI group. Moreover, the repair of colonic mucosal injury was observed by HE staining in the LOW group. It should be noted that these results in the MID group were more similar to the LOW group. Thus, low-dose sucrose has been identified to alleviate the symptoms of colitis, slow down colonic inflammation, and repair mucosal damage in mice.

A fundamental role of the gut microbe in the intestinal homeostasis of the host, especially in IBD patients, has been confirmed in recent years [46]. In almost all IBD models, the disease will be blocked or significantly relieved under germ-free conditions, indicating the crucial role of microorganisms in the development of IBD inflammation [47]. Studies have shown that the diversity of gut microbiota is reduced in IBD patients [48]. Consistent with these findings, our data showed that DSS treatment and high-dose sucrose intake reduced intestinal microbial richness and diversity in mice, which was reversed by low-dose sucrose administration. The intestinal microbiome of colitis mice is often in a disordered state, characterized by a significant decrease in *Bacteroides* and *Bifidobacterium* [49]. The correction of intestinal flora disorder can effectively alleviate IBD. A previous study has suggested that intake of simple sugars at 10% predisposed to colitis and enhanced its pathogenesis via modulation of gut microbiota in mice [50]. Our study showed that different doses of sucrose significantly changed the composition of intestinal microbiota. The abundance of probiotic bacteria *Bacteroidetes*, *Acidobacteria*, and *Actinobacteria* increased in the LOW group compared to the DSS group. Meanwhile, the dominant bacteria in the HI group were *Lachnospiraceae_NK4A136_group*, *Ruminococcaceae*, and *Rhodospirillales*. Additionally, LEfSe analysis showed that the abundance of *Erysipelotrichaceae*, *Bifidobacterium*, *Faecalibaculum*, *Bacteroides*, and *Romboutsia* were much higher in the LOW group than the DSS group, which aroused our strong interest.

Several studies have revealed the potential ability *of Erysipelotrichaceae*, *Bifidobacterium*, *Faecalibaculum*, *Bacteroides*, and *Romboutsia* to produce SCFAs [51,52]. SCFAs, composed of acetic acid, propionic acid, butyric acid, etc., have been identified as cardinal examples of beneficial metabolites [53,54]. Reconstitution of Apc^Min/+^ or azoxymethane and dextran sulfate sodium-treated mice with an isolate of *Faecalibaculum*, which releases SCFAs, especially butyric, can reduce tumor growth [55]. A previous study has linked butyrate to commensal microbe-mediated induction of functional Treg cells [56]. A significant loss in the bacteria that produce SCFAs, such as propionate, is also a consequence of dysbiosis in CD patients [57]. The process involved in the production of SCFAs are associated with complex enzymatic pathways, accompanied by the activity of the glycolysis pathway and carbohydrate metabolism [58]. In line with previous studies, the PICRUSt analysis of the functional gene difference of the microbial community demonstrated that the starch and sucrose metabolism pathway and glycolysis/gluconeogenesis pathways were enriched in the LOW group. Indeed, the contents of SCFAs in colonic contents were markedly increased in the LOW group, consistent with gut microbiota composition in our study. Interestingly, the MID group and HI group showed similar results on the contents of SCFAs. Because of their similar composition of gut microbe, the key role of gut microbiota in the production of SCFAs has been confirmed.

PPAR-γ, a member of the nuclear receptor family of transcription factors, can inhibit the production of monocyte inflammatory cytokines [59] and promote the differentiation of Treg cells [60]. A recent study reports that Barley leaf supplementation attenuates colitis and resulted in the enrichment of microbiota-derived purine metabolite inosine, which could activate PPAR-γ signaling [61]. Moreover, evidence suggested that PPAR-γ activated by butyrate could precipitate colonocytes preferring β-oxidation, thereby inhibiting the expansion of pathogenic *Enterobacteriaceae* [34]. Therefore, the expression of PPAR-γ in the colon was detected to explore whether the high level of gut microbiota-derived SCFAs in the LOW group could activate PPAR-γ. The result showed that the level of PPAR-γ in the LOW group increased significantly as expected. 

PPAR-γ, as a ligand-activated transcription factor, has been implicated in the MAPK signaling pathway. MAPK cascades cooperate in the orchestration of inflammatory responses and extensive crosstalk to NF-κB [62]. The activation of PPAR-γ could alleviate colitis by effectively inhibiting NF-κB. Hence, the MAPK/NF-κB signaling pathway in DSS-induced colitis was measured to explore the remission mechanism of low-dose sucrose. The results showed that high-level PPAR-γ in the LOW group effectively inhibited the phosphorylation level of the MAPK/NF-κB signal pathway, which was consistent with the level of inflammatory cytokines. Moreover, the Spearman correlation analysis revealed that there was a strongly positive correlation between *Faecalibaculum* and the PPAR-γ as well the SCFAs levels. However, *Faecalibaculum* negatively correlated with the inflammatory factor levels in the colon. The IL-6 and IL-1β levels showed significantly negative correlations with *Bacteroides* and *Romboutsia*. Additionally, it is reported that *Lachnospiraceae_NK4A136* and *Ruminococcaceae* were enriched in colitis and positively correlated with the pathological feature [48,63]. In accordance with that, *Lachnospiraceae_NK4A136* and *Ruminococcaceae*, the specific biomarkers in the HI group, were negatively correlated with PPAR-γ and SCFAs levels, especially butyric acid levels. Together, we found that the amount of sucrose played a critical role in the development of the disease. Therefore, the impact of gut microbiota should be taken into account in the recommendations on the safe use of sugar intake. Undoubtedly, the mechanism of the effect of sucrose intake on colitis still needs to be further studied. Long-term and more sophisticated exposure experiments should be conducted both in healthy and susceptible populations.

## 5. Conclusions

In the present study, we demonstrated that different dosages of sucrose consumption altered the disease severity and microbial homeostasis differently in DSS-induced colitis in mice. In contrast to the high-dose sucrose, the low-dose sucrose intervention alleviated colitis in mice by suppressing colon inflammation and repairing mucosal injury. Mechanistically, low-dose sucrose intervention restored microbial dysfunction, promoted the production of SCFAs, and then activated the PPAR-γ/ MAPK/NF-κB signaling pathway. Our study provides a new perspective on the safe use of sucrose. It can be concluded that low-dose intake of sucrose may prevent the occurrence of colitis, whereas high-dose sucrose aggravates the symptoms of colitis. Above all, sugar is not the root of all evil; however, excess sugar is.

## Figures and Tables

**Figure 1 nutrients-14-02765-f001:**
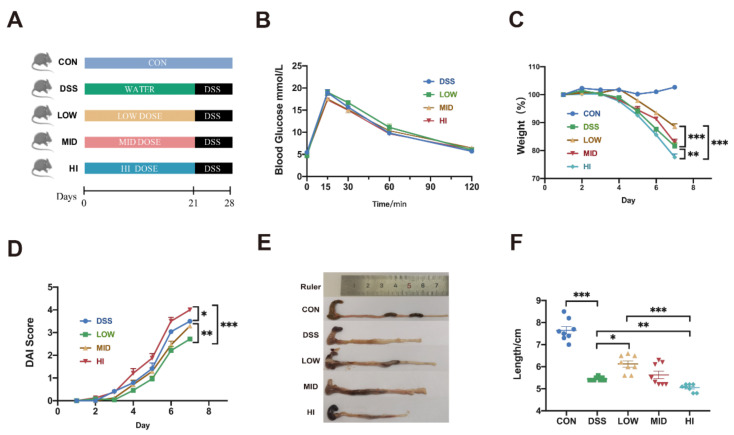
Effects of Different Dosage Sucrose on Symptoms of DSS-induced Colitis in Mice. (**A**) Schematic diagram of animal experiment design. (**B**) Oral glucose tolerance test (OGTT) after different doses of sucrose intervention for 3 weeks. (**C**) Alteration of body weight during DSS modeling. (**D**) Change of disease activity index. (**E**) Representative colonic photographs and colon length. (**F**) Colon length. The data were represented as mean ± SEM (*n* = 8). Analysis of weight and OGTT in experiments were performed by two-way ANOVA with Turkey post-hoc multiple comparisons. * *p* < 0.05, ** *p* < 0.01, *** *p* < 0.001.

**Figure 2 nutrients-14-02765-f002:**
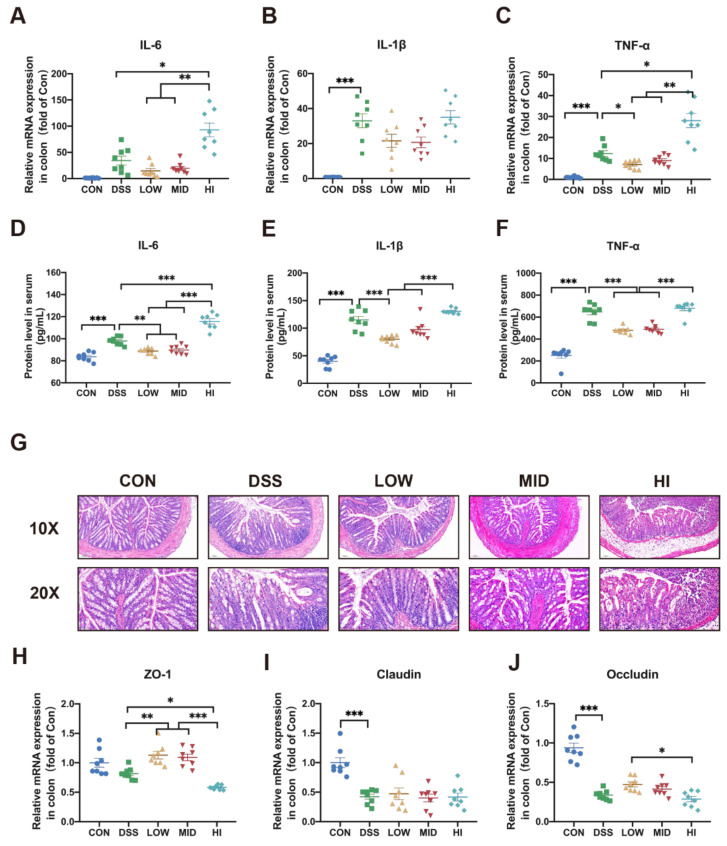
Effects of Different Dosage Sucrose on Colon Inflammation and Mucosal Barrier (**A**) The expression level of IL-6 in colon tissue. (**B**) The expression level of IL-1β in colon tissue. (**C**) The expression level of TNF-α in colon tissue. (**D**) The expression level of IL-6 in serum analyzed by ELISA. (**E**) The expression level of IL-1β in serum analyzed by ELISA. (**F**) The expression level of TNF-α in serum analyzed by ELISA. (**G**) Representative microscopy images (10× and 20× magnification, scale 100 μm for 10× and 50 μm for 20×) for H&E staining of colon (**H**) The expression level of ZO-1 in colon tissue. (I) The expression level of Claudin in colon tissue. (**J**) The expression level of Occludin in colon tissue. The data were represented as mean ± SEM (*n* = 8). * *p* < 0.05, ** *p* < 0.01, *** *p* < 0.001.

**Figure 3 nutrients-14-02765-f003:**
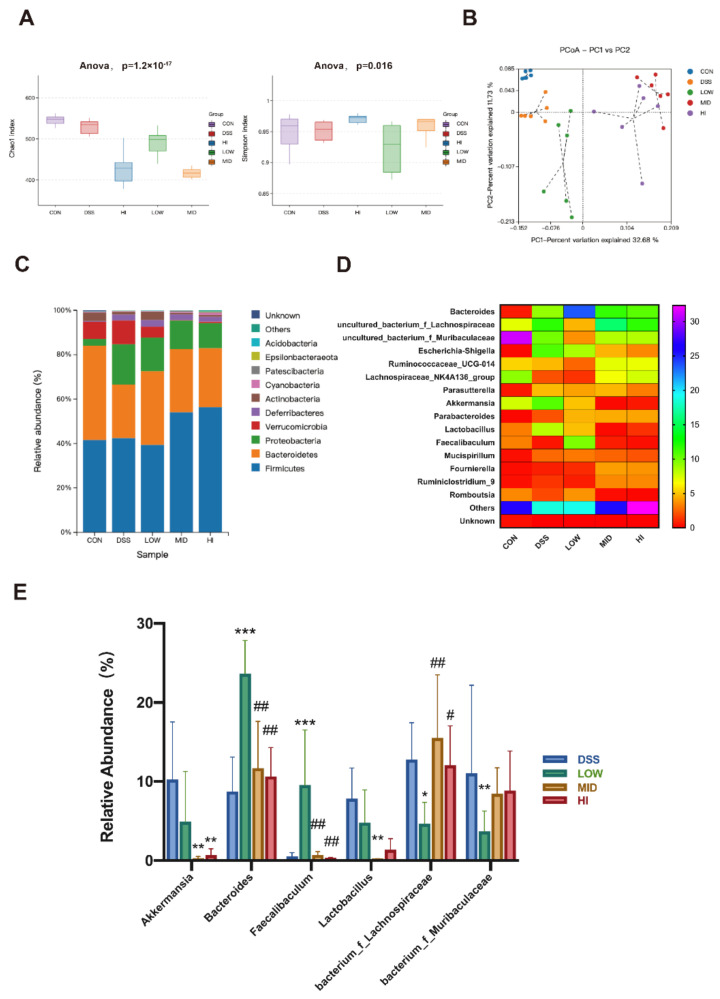
Effects of Different Dosage Sucrose on the Gut Microbiota Composition (**A**) The alpha diversity (Chao index and Simpson index). (**B**) The PCoA analysis of binary_jaccard distances for beta diversity. (**C**) The relative abundance of microbial at phylum level. (**D**) The relative abundance of microbial at genus level. (**E**) Representative species with significant alteration in abundance. The data were represented as mean ± SEM (*n* = 6–8). * *p* < 0.05 indicated a significant difference vs the DSS group. # *p* < 0.05 indicated a significant difference vs the Low group. * *p* < 0.05, ** *p* < 0.01, *** *p* < 0.001; # *p* < 0.05, ## *p* < 0.01.

**Figure 4 nutrients-14-02765-f004:**
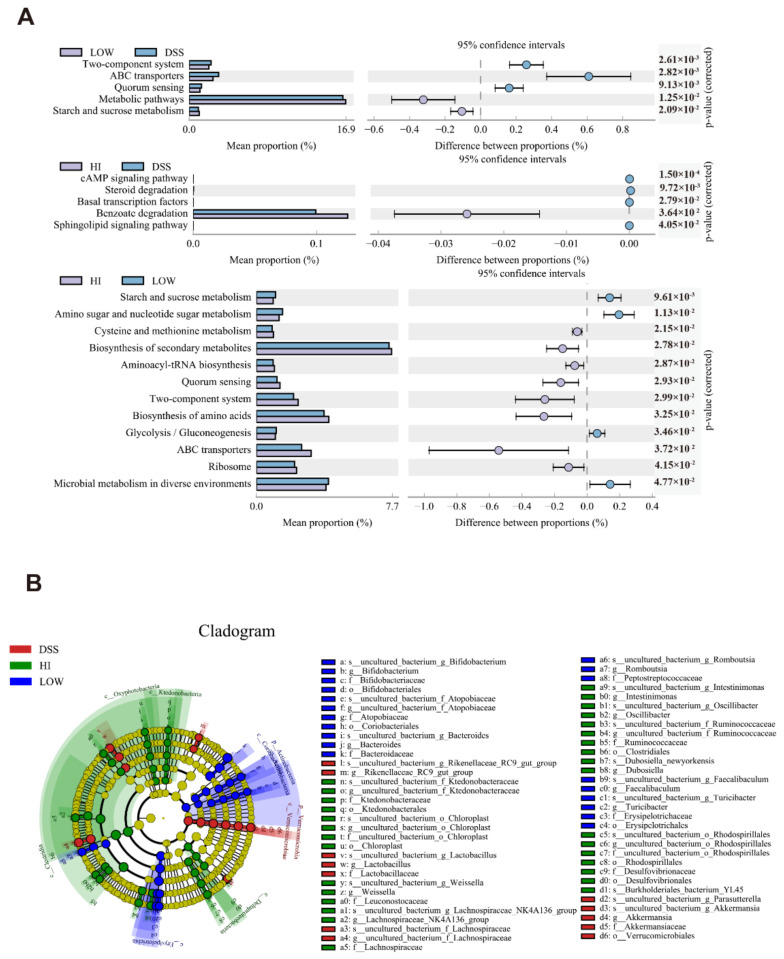
Effects of Different Dosage Sucrose on the Microbial Metabolism Pathways (**A**) PICRUSt analysis under the KEGG metabolic pathway (level 3) between different groups. (**B**) LEfSe analysis at the taxonomic level from phylum to genus. The data were represented as mean ± SEM (*n* = 6–8).

**Figure 5 nutrients-14-02765-f005:**
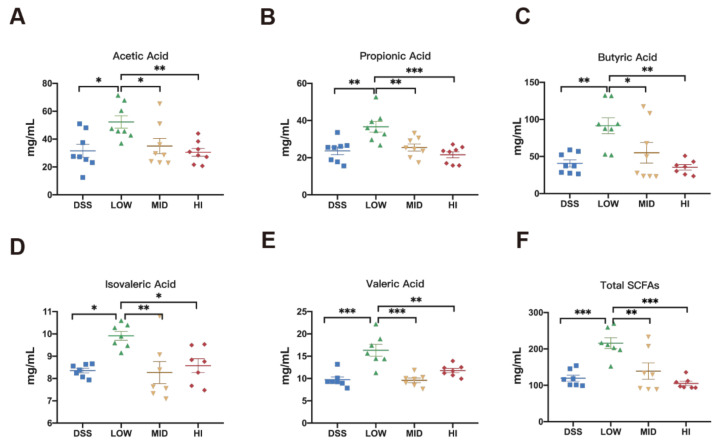
Effects of Different Dosage Sucrose on the SCFAs Production (**A**) The content of acetic acid. (**B**) The content of propionic acid. (**C**) The content of butyric acid. (**D**) The content of isovaleric acid. (**E**) The content of valeric acid. (**F**) The content of total SCFAs. The data were represented as mean ± SEM (*n* = 6–8). * *p* < 0.05, ** *p* < 0.01, *** *p* < 0.001.

**Figure 6 nutrients-14-02765-f006:**
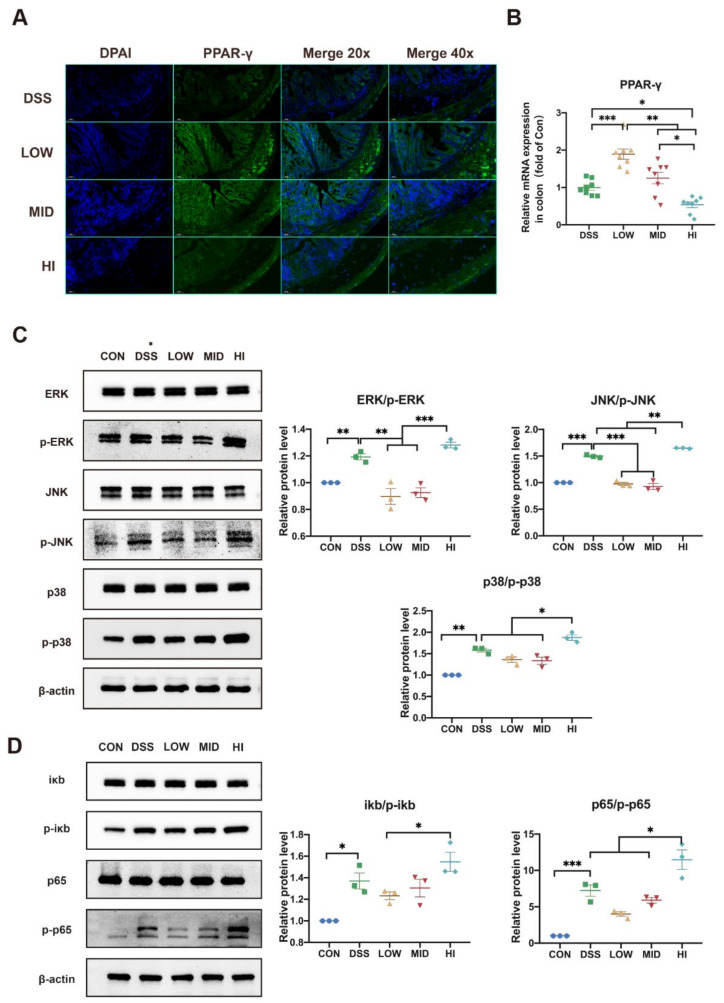
Effects of Different Dosage Sucrose on PPAR-γ and the MAPK/NF-κB Pathway. (**A**) Immunofluorescent analysis of PPAR-γ (green) in colonic sections (20× and 40× magnification, scale 100 μm for 20× and 50 μm for 40×). Nuclei were stained with DAPI (blue). (**B**) The expression level of PPAR-γ in colon tissue. (**C**) The total proteins and phosphorylated proteins of the MAPK pathway detected by Western blot. (**D**) The total proteins and phosphorylated proteins of the NF-κB pathway detected by Western blot. The data were represented as mean ± SEM (*n* = 3 or 8). * *p* < 0.05, ** *p* < 0.01, *** *p* < 0.001.

**Figure 7 nutrients-14-02765-f007:**
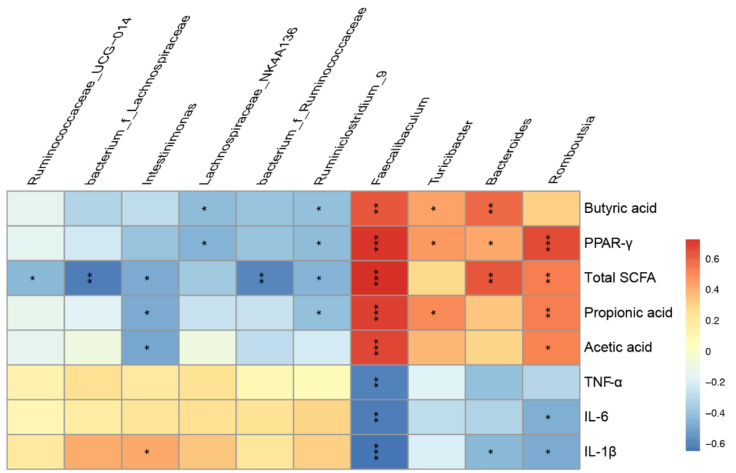
The Spearman correlation analysis at genus level. The data were represented as mean ± SEM (*n* = 6). * *p* < 0.05, ** *p* < 0.01, *** *p* < 0.001.

## Data Availability

The data presented in this study are available on request from the corresponding author.

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
