# Peer review of "Different Dose of Sucrose Consumption Divergently Influences Gut Microbiota and PPAR-γ/MAPK/NF-κB Pathway in DSS-Induced Colitis Mice"

_nutrients, 2022, doi:10.3390/nu14132765_

Round 1
Reviewer 1 Report
This manuscript entitled with "Divergently Regulation of Different Dose Sucrose Consumption on Gut Microbiota and PPAR-γ/MAPK/NF-κB Pathway in DSS-Induced Colitis Mice" conducted a study to characterize impact of different doses of sucrose consumption in DSS-induced colitis mice.
This manuscript was well-thought and may have a significant impact to the current research field. However, it has to be rejected to publish in Nutrients due to non-rigorous and insufficient execution.
Major:
- Figure 1: there is no unit for Fig1B; and no axis titles for Figs 1C and 1D. Figures without unit or titles were not readable and trustable.
- Figure 3: gut microbiota composition for the control group was missing.
- Figure 5: western blot without repeats and quantification was not acceptable.
- Seriously, the mechanism of sucrose on colitis was still not clear after reading, therefore it cannot be accepted with current simple experiments and insufficient explanation and discussion.
Minor:
- Figure 3: font size of figs 3A -3C were too small to read. Please keep the font size clear and consistent in each figure.
- Figure 4: font size of figs 4A-4B were too small to read.
- Figure 5: figure quality for fig 5A was not clear to reveal the difference.
Author Response
General comments:
This manuscript was well-thought and may have a significant impact to the current research field. However, it has to be rejected to publish in Nutrients due to non-rigorous and insufficient execution.
Response:
Thank you very much for your pertinent comments on this article. We regret that the initial manuscript did not meet your requirements. The manuscript has been re-organized and updated throughly according to your professional comments and suggestions. We sincerely hope that the revised manuscript can be approved by you. The manuscript has been carefully revised and the major revisions were marked in the revised manuscript. If there is something unclear, please don’t hesitate to contact us.
Major:
Comment 1
Figure 1: there is no unit for Fig1B; and no axis titles for Figs 1C and 1D. Figures without unit or titles were not readable and trustable.
Response
Special thanks for your careful correction. The associated Figure 1 has been revised in the manuscript. We have re-checked all the figures carefully to ensure the integrity and correctness of the titles and units.
Comment 2
Figure 3: gut microbiota composition for the control group was missing.
Response:
Thanks for your professional advice. Results of gut microbiota analysis on the control group have been loaded into the manuscript in Figure3A-D. As you suggested, the addition of the control group can intuitively show the changes in the gut microbiota composition of mice with different doses of sucrose and 2.5% DSS. Such results may better support our conclusion that different doses of sucrose intervention alter gut microbiota compositions in mice, resulting in different colitis symptoms after DSS challenge. It needs to be further explained that we did not analyze the control group in the other analysis of gut microbiota, because we hoped to be able to compare the differences between the groups that received the DSS challenge to reflect the influence of different doses of sucrose to colitis.
Comment 3
Figure 5: western blot without repeats and quantification was not acceptable.
Response:
Thanks for your comment. As you suggested, quantification and repetition of western blot are required to ensure reliable results. In fact, we performed three replicates for each group, and the relevant results have been provided in the supplementary material. In addition, we performed intensity measurements of the bands using ImageJ. The relevant results have been updated in Figure 6C-D of the manuscript.
Comment 4
Seriously, the mechanism of sucrose on colitis was still not clear after reading, therefore it cannot be accepted with current simple experiments and insufficient explanation and discussion.
Response:
Thank you for your technical review. In this study, influences of three doses of sucrose on weight, diarrhea, colon histomorphology, inflammatory cytokines, gut microbial composition and SCFAs production changes in DSS-induced colitis mice were evaluated systematically. We demonstrated that different doses of sucrose intake altered the disease severity and microbial homeostasis differently. Of note, low dose sucrose intake relieved the colonic inflammation and amended the gut dysbiosis. Meanwhile, excess sucrose consumption disturbed microbial homeostasis and exacerbated the inflammation. Mechanistically, the enhancement of PPAR-γ related signaling pathways activated by SCFAs involved in the remission of colitis. As you mentioned, further experiments may need to be initiated. The relationship between microbes and sucrose intake remains open to exploration. Germ-free mice or fecal microbiota transplantation assays may be utilized for further exploration. Furthermore, consistent with previous studies, our study suggests that the relationship between PPAR-γ and microorganisms is coupled through SCFAs. However, whether sucrose is a direct substrate for SCFAs production, or sucrose affects SCFAs production through cross-feeding, these are all things that we can study further. Thanks again for your careful review and inspiration for our future research.
Minor:
Comment 5
Figure 3: font size of figs 3A -3C were too small to read. Please keep the font size clear and consistent in each figure.
Response:
Thanks for your kind suggestion. We have rearranged the pictures. The problem you mentioned has been modified in the Figure 3. Sorry for the inconvenience in your reading. Higher resolution images have been replaced to ensure accessibility of the results.
Comment 6
Figure 4: font size of figs 4A-4B were too small to read.
Response:
We are very grateful to your comment. The Figure4A-4B have been rearranged and adjusted for easier reading.
Comment 7
Figure 5: figure quality for fig 5A was not clear to reveal the difference.
Response:
Thanks for your careful evaluation. We've replaced it with sharper figure to ensure readers can better identify the results, you can check the relevant modifications in Figure 6.
Reviewer 2 Report
Studies on basic mechanisms are really appreciated as they increase knowledge about the interactions among diet, microbiota and the host. This study by Zhang and colleagues has also this potential. Nevertheless, there are several issues that need to be addressed:
Overall, English language should be improved as some of the sentences are missing sense. This could markedly improve readability of the manuscript.
The references 11 and 12 are not referring to mouse studies. You can add some, such as Fajstova et al. 2020, and this particular should be also discussed.
In the section 2.1., you write that the groups consist of 10 mice each but on figures and in figure legends, there are mostly results from only 8 mice. Why?
You are using Prism version 8.0 for statistical analyses which is fine but it does not include any adjustment of p value for multiple comparison in correlation analyses. Did you adjust your data on Figure 5E? If not please recalculate your data.
Can you add control group into gut microbiota analyses? It is important to see the background microbita to clearly identify the shifts after dietary intervention.
The differences in data measured by Western blot are not clearly visible by eye. To improve it, you can add band intensity measurement, to show relative protein expression.
Author Response
Comment 1
Overall, English language should be improved as some of the sentences are missing sense. This could markedly improve readability of the manuscript.
Response:Thank you for your reminder. We have revised several grammar problems as you suggested. Meanwhile, the revised manuscript has been thoroughly reviewed and checked by co-authors to minimize errors in language. All the mistakes have been revised and marked in the revised manuscript.
Comment 2
The references 11 and 12 are not referring to mouse studies. You can add some, such as Fajstova et al. 2020, and this particular should be also discussed.
Response:
We are very grateful for your professional advice. The article you mentioned has been cited in the manuscript. In order to get a more profound point of view, we carefully read this article. Also, a related discussion can be seen at Lines 352.
Comment 3
In the section 2.1., you write that the groups consist of 10 mice each but on figures and in figure legends, there are mostly results from only 8 mice. Why?
Response:
Thanks for your comment. 10 mice were utilized in all our experiments. When performing statistics and analysis of the results, we excluded discrete points and retained as much centralized data as possible. Furthermore, to make the data analysis method more powerful, we chose to keep the same number of results and the significances of different groups were performed by one-way ANOVA followed by Tukey's post hoc test. We believe that this result can not only ensure the centrality of the data, but also show the overall effect.
Comment 4
You are using Prism version 8.0 for statistical analyses which is fine but it does not include any adjustment of p value for multiple comparison in correlation analyses. Did you adjust your data on Figure 5E? If not please recalculate your data.
Response:
We are very grateful to your comment. The correlation heat map is obtained through the microbial diversity analysis platform provided by BMKCloud. Spearman analysis was used to calculate the correlation, and the multiple comparison in correlation analyses were tested using the function corr.test provided by the psych package, and finally plotted using pheatmap_1.0.12.
Comment 5
Can you add control group into gut microbiota analyses? It is important to see the background microbita to clearly identify the shifts after dietary intervention.
Response:
Results of gut microbiota analysis on the control group have been loaded into the manuscript and Figure3A-D. As you suggested, the addition of the control group can intuitively show the changes in the gut microbiota composition of mice with different doses of sucrose and 2.5% DSS. Such results may better support our conclusion that different doses of sucrose intervention alter gut microbiota compositions in mice, resulting in different colitis symptoms after DSS challenge.
Comment 6
The differences in data measured by Western blot are not clearly visible by eye. To improve it, you can add band intensity measurement, to show relative protein expression.
Response:
Thank you for your technical review. We have performed intensity measurements of the bands using ImageJ. The relevant results have been updated in Figure 6C-D.
Reviewer 3 Report
The article presented by Xuejiao Zhang and collaborates, entitled “Divergently Regulation of Different Dose Sucrose Consumption on Gut Microbiota and PPAR-γ/MAPK/NF-κB Pathway in DSS-Induced Colitis Mice”, is an original article that aimed to elucidate the potential role of three doses of sucrose on DSS- induced colitis in C57BL/6 mice and the underlying mechanisms.
From what can be deduced from the material and methods, the different amounts of sugar were administered before the treatment with DSS, and once the treatment with DSS began, the administration of sugar was suspended. All this indicates, therefore, that it is a model that works on sugar levels that help prevent disease. This should be better reflected throughout the work. Since the model does not speak of intervention in the sugar diet in a mouse with already established disease, which is what usually occurs in patients with Ulcerative Colitis.
Major revision:
1. Line 74: At the end of the introduction it is necessary to include a working hypothesis together with the objective.
Minor revision
1. Line 21: SCFAs. it is necessary to specify the meaning of the acronyms in the abstract
2. Line 60: SCFAs. it is necessary to specify the meaning of the acronyms in the introduction
3. Line 85, 342: 25g, separate
4. Line 95: DAI score. it is necessary to specify the meaning of the acronyms and the score
5. Line 99: 1-2cm, separate
6. Line 121: 30min, separate
7. Figure 2 G and 5A. Scale
8. Molecular weight Figure 5C. Quantification WB. The authors must indicate the machine used to obtain the western blot images and the program used for the quantification of the bands.
Author Response
Comment 1
Line 74: At the end of the introduction it is necessary to include a working hypothesis together with the objective.
Response:
Thank you very much for your professional advice. Based on your suggestion, the research purpose and our research hypotheses have been included in the introduction section at Lines 75-83. In this study, influences of three doses of sucrose on weight, diarrhea, colon histomorphology, inflammatory cytokines, gut microbial composition and SCFA production changes in DSS-induced colitis mice were evaluated systematically. We demonstrated that different doses of sucrose intake altered the disease severity and microbial homeostasis differently. Of note, low dose sucrose intake relieved the colonic inflammation and amended the gut dysbiosis. Meanwhile, excess sucrose consumption disturbed microbial homeostasis and exacerbated the inflammation. Mechanistically, the enhancement of PPAR-γ related signaling pathways activated by SCFAs involved in the remission of colitis. Thanks again for your suggestion, which makes our introduction more scientific and rigorous.
Comment 2
Line 21: SCFAs. it is necessary to specify the meaning of the acronyms in the abstract
Response:
Thanks for your kind suggestion. Following your suggestion, we have specified the meaning of the acronyms in the abstract. The manuscript was checked to ensure that the same errors did not occur. You can find the relevant content revised in Lines 21 of the manuscript.
Comment 3
Line 60: SCFAs. it is necessary to specify the meaning of the acronyms in the introduction
Response:
Thanks for your careful evaluation. According to your suggestion, we have corrected the relevant content in the manuscript.
Comment 4
Line 85, 342: 25g, separate
Response:
Thanks for your careful inspection, we have revised the format at Lines 95 and 358.
Comment 5
Line 95: DAI score. it is necessary to specify the meaning of the acronyms and the score
Response:
We appreciate your suggestion. The disease activity index (DAI) of every experimental animal was monitored simultaneously (Supplementary Table S1).
Supplementary Table S1. Disease Activity Index Score
Disease Activity Index Score |
|||
score |
Rectal bleeding |
Stool consistency |
Weight loss(%) |
0 |
none |
normal |
0 |
1 |
occult blood negative |
soft but formed |
1-5 |
2 |
occult blood positive slightly |
loose |
6-10 |
3 |
occult blood positive |
mild diarrhea |
11-15 |
4 |
gross bleeding |
severe diarrhea |
>15 |
The final macroscopic score for each animal is the average of these three separate scores.
Comment 6
Line 99: 1-2cm, separate
Response:
Thanks for your careful inspection, we have revised the format at Lines 108.
Comment 7
Line 121: 30min, separate
Response:
Thanks for your careful inspection, we have revised the format at Lines 130.
Comment 8
Figure 2 G and 5A. Scale
Response:
Thanks for your careful inspection. The scale has been defined in the legend of the Figure 2 G and 5A.
Comment 9
Molecular weight Figure 5C. Quantification WB. The authors must indicate the machine used to obtain the western blot images and the program used for the quantification of the bands.
Response:
We are very grateful to your comments. The molecular weight has been defined in the Materials and Methods chapter 2.6 Western blotting. We have performed intensity measurements of the bands using ImageJ. The Bio-Rad ChemiDocTM MP Imageing System was used for image acquisition. Following your suggestion, the relevant content has been updated in the manuscript in Lines 142-148,152-153.
Round 2
Reviewer 1 Report
The authors have revised according to reviewer's comment, please update Figure 4B with larger and visible font size.
The manuscript can be accepted after minor revision.
Author Response
Thanks for your careful checks. According to the size limit of the figure, we have enlarged the font in the figure as much as possible. However, if it still does not meet your requirements, please do not hesitate to let us know.
Reviewer 3 Report
The authors have answered my questions
Author Response
Special thanks for your constructive comments, we have learned a lot from them.